# Does the Type of Knee Arthroplasty Affect the Patient’s Postural Stability?

**DOI:** 10.3390/medicina60101582

**Published:** 2024-09-26

**Authors:** Tomasz Sorysz, Aleksandra Adamik, Katarzyna Ogrodzka-Ciechanowicz

**Affiliations:** 1Trauma and Orthopaedic Unit, Gabriel Narutowicz Municipal Specialist Hospital in Krakow, 31-202 Krakow, Poland; tsorysz@gmail.com; 2Institute of Applied Sciences, Faculty of Motor Rehabilitation, University of Physical Education, 31-571 Krakow, Poland; ola.adamik99@gmail.com; 3Institute of Clinical Rehabilitation, Faculty of Motor Rehabilitation, University of Physical Education, 31-571 Krakow, Poland

**Keywords:** stabilometry, knee arthroplasty, postural stability

## Abstract

*Background and Objectives:* The aim of this study was to assess postural stability in patients after total and unicompartmental knee arthroplasties. *Materials and Methods*: The study included 40 women who had undergone knee arthroplasties—20 women who had undergone total knee arthroplasty (TKA) (mean age 63.47 ± 2.17) and 20 women who had undergone unicompartmental knee arthroplasty (UKA) (mean age 64.65 ± 1.93). The comparison group consisted of 20 healthy women aged 60–69 years (mean age 64.45 ± 3.12). The average time from surgery to stabilometry was 14.4 months. Each patient underwent stabilography using a single-plate stabilography platform, which included both Romberg’s test and a dynamic test. Additionally, the WOMAC scale was administered, where patients assessed their condition both before surgery and at the present time. *Results*: The averaged Romberg’s test results show a slight displacement in the center of mass (COM) toward the forefoot and towards the right limb in both the TKA and UKA groups. The WOMAC scale results showed significant improvement and satisfactory functional outcomes in both groups. *Conclusions:* The study indicated that one year after surgery, patients in both groups required a larger base of support to maintain postural control. However, the results for the UKA group were more similar to those of healthy individuals.

## 1. Introduction

Maintaining postural stability is the result of proper neuromuscular coordination, which allows for the position of the center of gravity of the head over the pelvis and the projection of the general center of gravity onto the plane of support to be maintained [1]. From a biomechanical perspective, balance is the state in which the sum of forces acting on the body is zero. According to this assumption, the stabilizing internal forces should be equal to the destabilizing external forces. Thus, postural stability refers to the body’s resistance to disturbances in balance [2]. Blaszczyk et al. define postural stability as continuous motor control along with the maintenance of posture in response to disturbances [3].

Achieving a stable upright posture is an interaction between sensory inputs from proprioception, the visual system, vestibular system, motor system, and cognitive elements [1,4]. The visual system is responsible for visual control and locomotion, while the vestibular system allows for, among other things, the orientation of the body in space. These systems provide information about the head’s position relative to the rest of the body. The sensory system (mechanoreceptors and proprioceptors) is responsible for the positioning of body parts relative to each other and the environment [5].

The human body is unconsciously able to oscillate within a radius of 5 mm. If a stimulus disrupts balance, a corrective programme is required, which is an integral part of the movement programme. The area within which corrections can be implemented is known as the safety margin. If this margin is exceeded, it becomes impossible to regain balance, which can lead to a fall [6].

Progressive osteoarthritis of the knee causes joint pain, restricted joint mobility, postural stability problems, and ultimately leads to varying degrees of disability [7,8]. Knee arthroplasty alleviates the pain associated with knee osteoarthritis, corrects deformities, and improves the patient’s functional ability [9,10]. 

Depending on the extent of the disease, surgical treatment may involve unicompartmental (partial) or total knee arthroplasty.

The primary goal after knee arthroplasty is to achieve not only pain-free joint mobility but also a high level of activity and quality of life. Quality of life depends on the preoperative condition, comorbidities, socioeconomic and cultural factors, and various physical parameters, including balance and postural control [11]. Some studies have shown that up to 30% of patients are dissatisfied five years after surgery [12]. At the same time, only a few studies have reported a higher incidence of falls postoperatively compared to asymptomatic healthy older adults, ranging from 17% to 48% [11,13].

Stabilometry is the objective study of body sway during quiet standing, i.e., stance in the absence of any voluntary movements or external perturbations. Stabilometry is usually based on the analysis of the time variant center of pressure (CoP) coordinates during a bipedal or single leg stance with the eyes open or closed [14].

Postural stability is a critical element in controlling body position and maintaining static and dynamic balance during activities such as walking, getting out of bed, getting in and out of the bath, and performing household chores. Static balance is maintaining equilibrium when stationary, while dynamic balance is maintaining equilibrium when moving [15]. Impaired postural stability is one of the main causes of falls in older adults and, therefore, represents a significant public health issue in patients with osteoarthritis [16].

Impairment of postural stability develops due to proprioceptive deficits, muscle weakness, and knee pain in patients with knee osteoarthritis [16]. The relationship between trunk balance and postural stability parameters after total knee arthroplasty has been the subject of consideration by researchers [17,18]. 

It is expected that postoperatively, the frequency of falls will decrease with the relief from pain, satisfactory function, and improvement in proprioception [19,20]. However, some studies have shown that the sacrifice of the anterior cruciate ligament (ACL) leads to proprioceptive/balance deficits after total knee arthroplasty, resulting in an increased risk of falls [21]. Moutzouri et al. have observed that patients after total knee arthroplasty (TKA) do not fully regain proprioception and quadriceps strength, which can be reduced by up to 60%, compromising symmetry and postural stability as a consequence [22]. 

Despite this, there are few evidence-based studies showing the results of stabilography indicators related to changes in body posture control after knee arthroplasty [17,23]. There is also a lack of such analyses conducted in patients with unicompartmental knee arthroplasty (UKA).

The aim of this study was to assess postural stability in patients after total and unicompartmental knee arthroplasty.

## 2. Materials and Methods

### 2.1. Study Design

This observational study was conducted in compliance with the Strengthening the Reporting of Observational Studies in Epidemiology (STROBE) Statement: guidelines for reporting observational studies and the ethical standards of the Committee on Human Experimentation of the institution where the experiments were adhered to or were in accordance with the Declaration of Helsinki of 1964 and its later amendments. This study received approval from the Bioethics Committee [32/KBL/OIL/2024] and all patients gave their written informed consent [24]. 

### 2.2. Participants

A total of 45 women from the Trauma and Orthopedic Department of the Gabriel Narutowicz Municipal Specialist Hospital in Kraków, who underwent TKA (23 women) or implantation of UKA (22 women), qualified for this study. 

Indications for the procedure in all patients were osteoarthritic changes in the knee joint. The average time from surgery to stabilometry was 14.4 months (14.4 ± 2.28). In all study subjects, an anterior midline incision with medial parapatellar approaches was performed.

Inclusion criteria: Undergoing TKA procedure or UKA implantation due to knee osteoarthritis;Absence of other diseases or injuries of the lower limbs;Absence of neurological conditions that could affect the study results;Ability to move independently after the surgery (without the use of orthopedic aids).

Exclusion criteria:
Inflammatory conditions and acute pain in the lower limbs;Lack of written consent from the patient to participate in the study.

The comparison group consisted of 20 healthy women aged 60–69 (mean age 64.45 ± 3.12), with no injuries or conditions affecting the lower limbs.

### 2.3. Setting

The study was conducted from January to April 2024 at the Laboratory of Functional Diagnostics of the Central Scientific and Research Laboratory (CLNB) at the University School of Physical Education in Kraków.

### 2.4. Outcome Measures

Each patient underwent stabilography using a single-plate stabilography platform. Additionally, the WOMAC scale was administered, where patients assessed their condition both before surgery and at the present time.

The study included:

Measurements of stabilometric indicators using the ALFA stabilography platform (AC International East Sp. z o.o., Knurów, Poland). (Figure 1)

The analysis included the following indicators:
In the Romberg’s test:
○COM X—average deviation on the X-axis (cm); ○COM Y—average deviation on the Y-axis (cm); ○V X—average velocity X (cm/s); ○V Y—average velocity Y (cm/s); ○PL—path length of COM (cm);○SA—surface area (cm^2^).In the dynamic test:
○Time R-F—time to reach target right front (s); ○Time L-F—time to reach target left front (s);○Time L-R—time to reach target left rear (s);○Time R-R—time to reach target right rear (s); ○PL—path length (cm).

Tests were conducted using the ALFA stabilometric platform.

The tests consisted of static and dynamic balance assessments. Stabilometric measurements were performed on barefoot participants (without socks). 

Static tests record changes in the position of the center of mass (COM) on the support plane in the transverse plane, specifying the directions of force: anterior–posterior and lateral left–right. 

In this study, the static tests were performed with both eyes open and closed. The dynamic test, utilizing visual feedback, required the participant to move their center of gravity (COG) while standing to the maximum possible deviation of the COG from the central point of the base of support. 

Each participant completed two tests—one with eyes open and one with eyes closed (Romberg’s test) and one dynamic test. Each patient underwent three trials of each test, from which the average result was calculated. The duration of each test was 30 s, with a 5-min break between trials. Height and weight measurements were taken before the tests.

During the static balance test with eyes open, the participant stood freely (barefoot) on the platform in an upright position. The participant’s feet were placed hip-width apart, with their arms hanging loosely by the sides and eyes open, focusing on a monitor positioned at eye level and placed 1 m from them.

Each participant stood on the platform in a relaxed position, with their feet parallel and a 10-cm distance measured from the head of the first metatarsal to the central line of the platform. The lateral malleoli were positioned on a perpendicular line dividing the platform in half, running 15 cm from the rear edge of the platform. The signal registration from the platform was synchronized in time with the moving point on the screen. The point on the screen moved according to the direction and range of body stability. The test result was the length of the deviation path caused by the ground reaction force due to foot pressure. Deviations to the left and backward were represented with a negative sign, while deviations to the right and forward were represented with a positive sign. 

Conditions for the static balance test with eyes closed were similar to those for the static balance test with eyes open, but the participants eyes were closed. 

The conditions for conducting the dynamic test remain unchanged from the previous tests. This test utilized visual feedback (the principle of biofeedback). The participant’s task was to hit as many points visible on the screen as possible by balancing the body in the shortest amount of time without lifting their feet off the ground. The points lit up in yellow at random. Hitting a point was confirmed by an audible signal. The test was conducted at an easy level. The test result was the length of the boundary path of body stability (COG), caused by the ground reaction force due to foot pressure, expressed in centimeters, and the time it took for the center of foot pressure (COP) to move.

Each participant was instructed about the test procedure beforehand. During the tests with eyes open, the patient was asked to focus on a specific point. During the tests with eyes closed and dynamic tests, the participant was assisted. The tests were conducted in silence, allowing the participant to concentrate on the task at hand. (Figure 2)

The comparison group of healthy individuals underwent the same tests in the same manner. 

#### Clinical Examination

The WOMAC scale (Western Ontario and McMaster Universities Index of Osteoarthritis) covers three areas of concern for patients: pain, disability in daily activities, and joint stiffness. It consists of 24 questions evaluated on a five-point scale. The patient responds by selecting one of five options (scoring 0–4). The maximum score a patient can achieve is 96 points, and the minimum is 0 (pain severity—20, stiffness—8, physical functioning—68). The points obtained from individual parameters are then totaled. Then, they are divided by 96 and multiplied by 100. The resulting score is a percentage and indicates the degree of functional impairment. The higher the score, the less functional the patient is, which suggests more severe degenerative changes in the knee joint [25]. 

### 2.5. Statistical Analysis

For statistical analysis, the Emmeans v1.10.1 and nlme3.1-162 packages were used in the R v4.2.2 environment. A Generalized Least Squares (GLS) model was employed that allowed variable variance between groups. The post-hoc test (Tukey) was used to compare differences between groups. Statistical analysis of the WOMAC test results was performed using Statistica 13.3 software (StatSoft, Tulsa, OK, USA). Measurement results were analyzed using descriptive statistical methods. Due to the nature of the variables analyzed, after verifying the normality of the distribution (Shapiro–Wilk test), a parametric Student’s *t*-test for dependent and independent samples was applied. Statistical significance was set at 0.01.

## 3. Results

Initially, 45 patients aged 55–70 who underwent TKA (23 women) or implantation of UKA (22 women) were enrolled in the study. After meeting the inclusion criteria, 20 patients after TKA (with a mean age of 63.47 ± 2.17) and 20 patients after UKA implantation (with a mean age of 64.65 ± 1.93) participated in the study. The comparison group consisted of 20 healthy women aged 60–69 years (mean age 64.45 ± 3.12). Patients agreed to participate in the study.

Table 1 contains detailed anthropometric data for the three groups. The qualification stage is presented in Figure 3.

The results from the Romberg test indicate significant differences between the study groups across all analyzed indicators, except for the comparison of the TKA and UKA groups (*p* = 0.15). The results clearly show that the patients in the UKA group achieved results similar to those of healthy individuals (see Table 2 and Table 3, Figure 4 and Figure 5).

The dynamic test also showed that stabilometric indicator measurement results differ significantly between the groups. Patients in the TKA group had a significantly weaker center of pressure (COP) movement time compared to both the UKA group and the group of healthy subjects. Similarly, the path length was the longest in the TKA group (182.59 cm) (see Table 4, Figure 6).

Comparison of WOMAC scale results in the TKA and UKA groups before and after surgery indicates a significant improvement. Patients clearly reported that their functionality significantly improved after surgery. The comparison of results between the groups after surgery is no longer statistically significant (*p* = 0.75) (Table 5).

## 4. Discussion

The aim of the study was to assess the static stability of patients after total and unicompartmental knee arthroplasty and to compare the results with those of healthy individuals. 

The study used a single-plate stabilography platform and the WOMAC scale, which enabled the assessment of the functional limitations of the patients. The averaged Romberg’s test results show a slight displacement of the center of mass (COM) toward the forefoot and towards the right limb in both the TKA and UKA groups. This trend may also be related to changes in body posture due to ageing and degenerative processes [26,27]. The results significantly differ from those of healthy individuals; however, the UKA group achieved results more similar to those of healthy individuals. It can be assumed that patients who underwent unicompartmental knee arthroplasty initially had less damaged joint surfaces than those who underwent total knee arthroplasty, indicating a less advanced disease process. However, it is important to remember that any disruption in knee stability due to osteoarthritis results in proprioceptive deficits, muscle weakness, and knee pain [16]. The patients in both groups were on average 14 months post-surgery, indicating that despite the surgery, postural control remains impaired, which may also increase the risk of falls. This could also be related to the progressive muscle weakness in older individuals; however, proprioceptive impairment in osteoarthritis is crucial. Davut et al. reached similar conclusions, assessing balance and fall risk in patients after TKA. They found that despite good functional outcomes, these patients’ dynamic balance was insufficient in the anterior, posterior, medial, and lateral directions [28]. 

Moutzouri et al. argue that TKA significantly improves single-leg standing balance and dynamic balance up to 1 year post-surgery. Furthermore, TKA positively impacts fear of falling and the frequency of falls. However, they emphasize that knee extension strength, proprioception, and postural stability are not fully restored after TKA, affecting balance performance [22]. These findings are consistent with the results of this study. 

It is also suggested that individuals with TKA require a wider base of support to manage surface modulation and more easily maintain dynamic stability. Comparisons with healthy individuals have shown that TKA does not fully restore static or dynamic postural control, and increased postural sway is still present 6 months post-surgery, not only on the prosthetic side but also on the contralateral side [29,30].

There is a lack of such comparisons in the available literature regarding patients with UKA. Goetz et al. note that the very good long-term outcomes and patient satisfaction with UKA are attributable to the preservation of both cruciate ligaments and bone tissue. However, despite better clinical and functional outcomes compared to TKA, UKA does not provide better proprioception than TKA [31]. Similar conclusions can be drawn from the results of this study. According to the WOMAC scale results in our study, patients in both groups reported significant improvement and satisfactory functional outcomes after surgery, but the stabilometry indicators were still worse than those of healthy individuals.

### Study Limitations

Like any study, this one had certain limitations. First, the study sample was small. However, attempts were made to minimize the bias and generalization of the data by applying various statistical tests. Further tests with a larger number of subjects are needed to be certain of these findings. Secondly, despite the significant conclusions that can be drawn from the results obtained, it would be beneficial to expand the study and conduct it both before and after TKA procedure and the implantation of UKA. Undoubtedly, the issue of knee alignment in TKA/UKA may also have an impact on postural stability. According to the available literature, there is an ongoing discussion on the effectiveness of kinematically aligned (KA) total knee arthroplasty (TKA) and mechanically aligned (MA) TKA. Experts indicate that this issue should be individualized and adapted to the capabilities of both the patient and the surgeons [32,33]. Nevertheless, when planning further studies on postural control, this aspect should also be taken into account.

In patients with OA with an indication for TKA, the results of the comparison of stabilometric indices indicate that the load is shifted to the healthy side, which directly indicates the unloading of the limb with OA [34]. It can therefore be assumed that after TKA/UKA this trend may persist and may have an impact on the recovery of postural control, but this requires further research.

## 5. Conclusions

Postural stability and control in patients after total and unicompartmental knee arthroplasty differ from those of healthy individuals.

The results indicate that 1 year after surgery, patients in both groups require a larger base of support to maintain postural control. However, the results for the UKA group were more similar to those of healthy individuals.

## Figures and Tables

**Figure 1 medicina-60-01582-f001:**
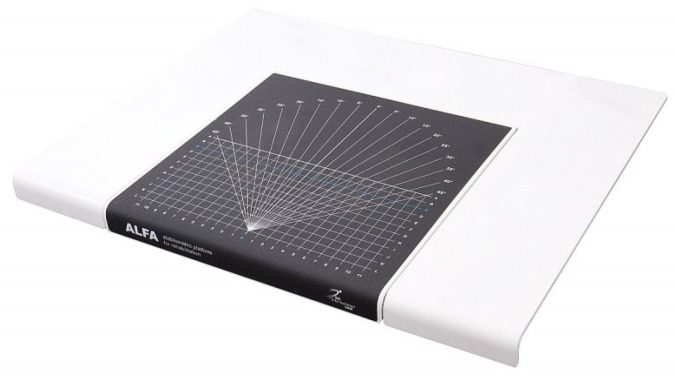
ALFA stabilography platform (AC International East).

**Figure 2 medicina-60-01582-f002:**
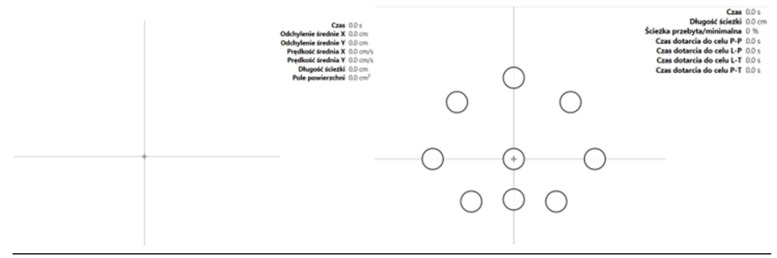
Image of static and dynamic test (own source).

**Figure 3 medicina-60-01582-f003:**
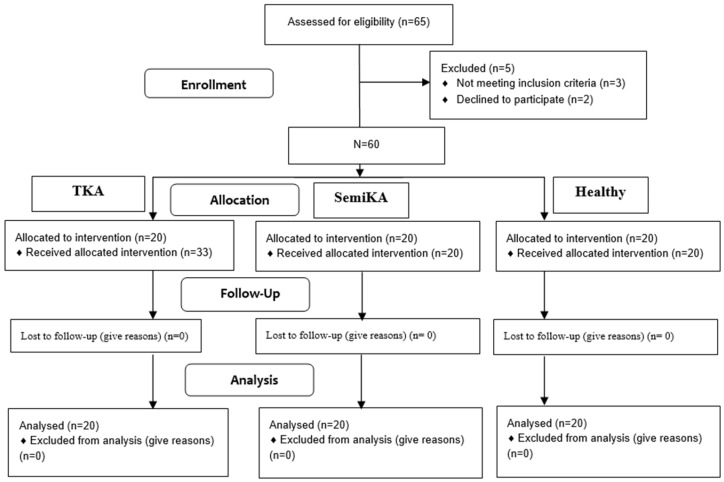
Flow diagram.

**Figure 4 medicina-60-01582-f004:**
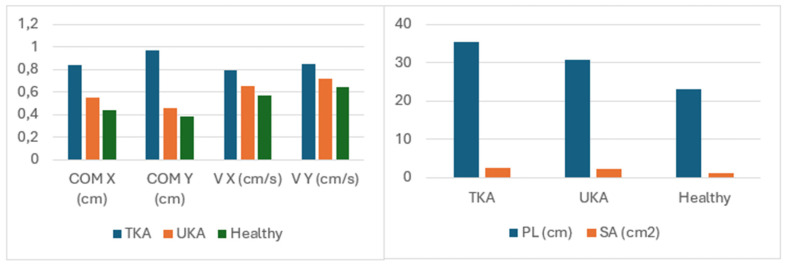
Comparison of stabilometric indicator measurement results (means) in the studied groups in the test with eyes open.

**Figure 5 medicina-60-01582-f005:**
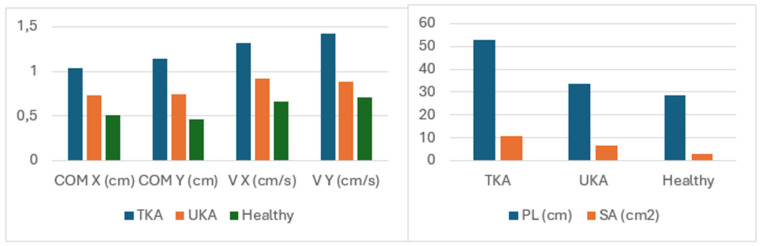
Comparison of stabilometric indicator measurement results (means) in the studied groups in the test with eyes closed.

**Figure 6 medicina-60-01582-f006:**
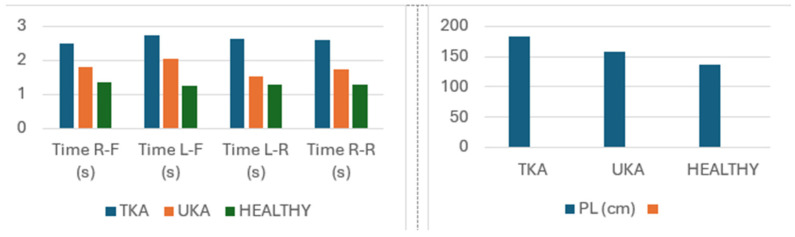
Comparison of stabilometric indicator measurement results (means) in the study groups in the dynamic test.

**Table 1 medicina-60-01582-t001:** Anthropometric data of the subjects in three groups.

Variable	TKA X ± SD; (Min/Max)	UKA X ± SD; (Min/Max)	Healthy X ± SD; (Min/Max)
Age [years]	63.47 ± 2.17 (59/69)	64.65 ± 1.93 60/70	64.45 ± 3.12 (60/69)
Height [cm]	164.81 ± 5.71 (158/179)	164.63 ± 4.61 (159/177)	165.45 ± 4.37 (159/181)
Body weight [kg]	77.11 ± 11.62 (59/108)	76.34 ± 8.32 (58/99)	74.20 ± 9.76 (61/101)

TKA—total knee replacement, UKA—single-compartment knee endoprosthesis; Healthy—comparison group; X ± SD – mean ± standard deviation.

**Table 2 medicina-60-01582-t002:** Comparison of stabilometric indicator measurement results in the studied groups in the test with eyes open.

Eyes Open				Group Comparison					
Indicators	TKA	UKA	Healthy	*t*			*p*		
	X ± SD	X ± SD	X ± SD	TKA–UKA	TKA–Healthy	UKA–Healthy	TKA–UKA	TKA–Healthy	UKA–Healthy
COM X (cm)	0.84 ± 0.33	0.55 ± 0.05	0.44 ± 0.02	3.87	5.32	8.80	<0.001	<0.001	<0.001
COM Y (cm)	0.97 ± 0.40	0.46 ± 0.05	0.38 ± 0.05	5.68	6.56	5.17	<0.001	<0.001	<0.001
V X (cm/s)	0.79 ± 0.14	0.65 ± 0.04	0.57 ± 0.04	4.50	6.98	6.79	<0.001	<0.001	<0.001
V Y (cm/s)	0.85 ± 0.11	0.72 ± 0.02	0.64 ± 0.03	5.10	7.89	9.07	<0.001	<0.001	<0.001
PL (cm)	35.37 ± 3.10	30.81 ± 0.82	23.06 ± 1.33	6.34	16.29	22.25	<0.001	<0.001	<0.001
SA (cm^2^)	2.51 ± 0.35	2.32 ± 0.31	1.24 ± 0.14	1.88	14.93	14.16	0.15	<0.001	<0.001

PL—path length; SA—surface area; COM X—average deviation on the X axis; COM Y—average deviation on the Y axis; V X—average velocity X; V Y—average velocity Y; TKA—total knee replacement, UKA—single-compartment knee endoprosthesis; Healthy—comparison group; *t*—Student’s *t*-test; *p*—level of significance.

**Table 3 medicina-60-01582-t003:** Comparison of stabilometric indicator measurement results in the studied groups in the test with eyes closed.

Eyes Closed				Group Comparison					
Indicators	TKA	UKA	Healthy	*t*			*p*		
	X ± SD	X ± SD	X ± SD	TKA–UKA	TKA–Healthy	UKA–Healthy	TKA–UKA	TKA–Healthy	UKA–Healthy
COM X (cm)	1.03 ± 0.34	0.73 ± 0.04	0.51 ± 0.02	3.83	6.66	21.86	<0.001	<0.001	<0.001
COM Y (cm)	1.14 ± 0.38	0.74 ± 0.04	0.46 ± 0.02	4.66	8.09	29.63	<0.001	<0.001	<0.001
V X (cm/s)	1.32 ± 0.21	0.92 ± 0.05	0.66 ± 0.03	8.45	14.26	19.28	<0.001	<0.001	<0.001
V Y (cm/s)	1.42 ± 0.18	0.88 ± 0.05	0.71 ± 0.03	12.92	17.30	13.66	<0.001	<0.001	<0.001
PL (cm)	52.72 ± 2.43	33.57 ± 1.54	28.72 ± 0.85	29.73	41.66	12.29	<0.001	<0.001	<0.001
SA (cm^2^)	10.55 ± 0.69	6.49 ± 0.63	2.87 ± 0.45	19.51	41.57	20.92	<0.001	<0.001	<0.001

PL—path length; SA—surface area; COM X—average deviation on the X axis; COM Y—average deviation on the Y axis; V X—average velocity X; V Y—average velocity Y; TKA—total knee replacement, UKA—single-compartment knee endoprosthesis; Healthy—comparison group; *t*—Student’s *t*-test; *p*—level of significance.

**Table 4 medicina-60-01582-t004:** Comparison of stabilometric indicator measurement results in the study groups in the dynamic test.

Dynamic Test				Group Comparison					
Indicators	TKA	UKA	Healthy	*t*			*p*		
	X ± SD	X ± SD	X ± SD	TKA–UKA	TKA–Healthy	UKA–Healthy	TKA–UKA	TKA–Healthy	UKA–Healthy
Time R-F (s)	2.49 ± 0.43	1.79 ± 0.10	1.34 ± 0.07	7.17	11.94	16.78	<0.001	<0.001	<0.001
Time L-F (s)	2.75 ± 0.37	2.03 ± 0.19	1.24 ± 0.06	7.79	17.98	17.87	<0.001	<0.001	<0.001
Time L-R (s)	2.63 ± 0.42	1.54 ± 0.05	1.29 ± 0.07	11.57	14.08	13.39	<0.001	<0.001	<0.001
Time R-R (s)	2.58 ± 0.39	1.72 ± 0.05	1.29 ± 0.06	9.73	14.54	22.77	<0.001	<0.001	<0.001
PL (cm)	182.59 ± 2.02	157.22 ± 2.76	135.96 ± 1.21	32.11	82.97	31.54	<0.001	<0.001	<0.001

Time R-F—time to reach target right front; Time L-F—time to reach target left front; Time L-R—time to reach target left rear; Time R-R—time to reach target right rear; PL—path length; TKA—total knee replacement, UKA—single-compartment knee endoprosthesis; Healthy—comparison group; X ± SD—mean± standard deviation; *t*—Student’s *t*-test; *p*—level of significance.

**Table 5 medicina-60-01582-t005:** WOMAC scale results.

WOMAC								
Group	Measurement	X ± SD	Me	Min–Max	*p*	Group Comparison	*p* before	*p* after
TKA	before	85.45 ± 4.19	84.50	79.00–92.00	0.001	TKA vs. UKA	0.03	0.75
	after	59.15 ± 10.77	59.00	42.00–80.00				
UKA	before	81.20 ± 7.57	82.50	65.00–92.00	0.001			
	after	58.05 ± 11.14	58.00	34.00–73.00				

before—before surgery, after—after surgery; TKA—total knee replacement, UKA—single-compartment knee endoprosthesis; X ± SD—mean ± standard deviation; Me—median; *p*—level of significance.

## Data Availability

The minimal data set is contained within our paper.

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
