# Peer review of "Does the Type of Knee Arthroplasty Affect the Patient’s Postural Stability?"

_medicina, 2024, doi:10.3390/medicina60101582_

Round 1
Reviewer 1 Report
Comments and Suggestions for Authors
Title : You should review the title. “ Knee endoprosthetis” isn’t an accurate and suitable term.
Line 23 : SemiKA. Replace this term with unicompartmental knee arthroplasty (UKA) in the whole paper.
One hand your paper is very interesting and fascinating, on the other hand there al some limitations :
· Type of TKA and UKA alignment : Nowdays you could treat a patient with knee OA
through different lower limb alignment ( mechanical, kinematical, functional, anatomical). You have clarify the TKA alignment, which could probably influence the postural stability.
DOI: 10.1016/j.arth.2023.01.060
· Knee approach : Surgical approach to the knee for TKA is pivotal. Many types of approaches are very invasive on the quadriceps, therefore they can alter its function. You should indicate this in the paper. Doi : 10.1016/j.jcot.2015.11.003
Table 1 : years > yers
Author Response
Thank you very much for your time and experience in providing feedback on the manuscript as well as valuable comments and suggestions regarding the article. Below is the answer to your review.
Title: You should review the title. “ Knee endoprosthetis” isn’t an accurate and suitable term.
Response: Thank you very much for pointing this out, the term "Knee endoprosthesis" has been changed to "Knee arthroplasty"
Line 23: SemiKA. Replace this term with unicompartmental knee arthroplasty (UKA) in the whole paper.
Response: Thank you very much for pointing this out, the term "SemiKA" has been changed to "UKA"
One hand your paper is very interesting and fascinating, on the other hand there al some limitations :
Type of TKA and UKA alignment: Nowdays you could treat a patient with knee OA through different lower limb alignment ( mechanical, kinematical, functional, anatomical). You have clarify the TKA alignment, which could probably influence the postural stability.
DOI: 10.1016/j.arth.2023.01.060
Response: Thank you very much for drawing attention to the problem of implant positioning and limb alignment. Undoubtedly, the issue of knee alignment in TKA/UKA may also have an impact on postural stability. According to the available literature, there is an ongoing discussion on the effectiveness of kinematically aligned (KA) total knee arthroplasty (TKA) and mechanically aligned (MA) TKA. Experts indicate that this issue should be individualized and adapted to the capabilities of both the patient and the surgeons. Nevertheless, when planning further studies on postural control, this aspect should also be taken into account. This information was included in the Study Limitation
Knee approach: Surgical approach to the knee for TKA is pivotal. Many types of approaches are very invasive on the quadriceps, therefore they can alter its function. You should indicate this in the paper. Doi : 10.1016/j.jcot.2015.11.003
Response: In all study subjects, anterior midline incision with medial parapatellar approaches was performed, which, according to the latest literature, results in better functional outcomes in the 3- and 6-month follow-up periods compared with the subvastus approach.
Table 1: years > yers
Response: the typo has been corrected
Reviewer 2 Report
Comments and Suggestions for Authors
First of all thank you for letting me review the manuscript entitled: “Does the type of knee endoprosthesis affect the patient's postural stability?”
On the other hand, the introduction is quite complete and easy for the reader to follow.
In material and methods it is not shown how the sample size was calculated or a sample power test. As for the results, it would be useful to integrate some graph to be able to see the results in a clearer way for the reader. The discussion is complete, although it could be further deepened in the comparison with other manuscripts such as the one published by this publisher. Implications of Stabilometric Assessment in Determining Functional Deficits in Patients with Severe Knee Osteoarthritis: Observational Study.
Congratulations for the work done.
Author Response
Thank you very much for your time and experience in providing feedback on the manuscript as well as valuable comments and suggestions regarding the article. Below is the answer to your review.
On the other hand, the introduction is quite complete and easy for the reader to follow.
1. In material and methods it is not shown how the sample size was calculated or a sample power test.
Response: The a priori assumption was three equal groups. Each group consisted of 20 people, so that statistical inference was reliable. The normal distribution was assessed (Shapiro-Wilk test) and further analysis was performed taking into account its results. The sample size also resulted from technical and organizational possibilities and the fact that the study was conducted on average 14 months after the procedure, so collecting a larger number of people was almost impossible.
2. As for the results, it would be useful to integrate some graph to be able to see the results in a clearer way for the reader.
Response: Figures have been added.
3. Discussion is complete, although it could be further deepened in the comparison with other manuscripts such as the one published by this publisher. Implications of Stabilometric Assessment in Determining Functional Deficits in Patients with Severe Knee Osteoarthritis: Observational Study.
Response: the proposed article refers to a comparison of stabilometric indices of patients with OA with an indication for TKA and healthy individuals. The results indicate that in patients with OA, the load was shifted to the healthy side, which directly indicates the relief and analgesic effect of the limb with OA. Our own studies refer to a comparison of patients after TKA and UKA and healthy individuals, and here a picture of anterolateral displacement of the COM is clearly drawn, which may increase the risk of falling.
In the Discussion, a fragment was added indicating the fact that in patients with OA with an indication for TKA, the results of the comparison of stabilometric indices indicate that in patients with OA, the load was shifted to the healthy side, which directly indicates the relief of the limb with OA.